# Distributed Nash Equilibrium Seeking for Multi-cluster UAVs Formation

1st Tao Hu
*College of Science, Liaoning University of Technology*
*name of organization (of Aff.)*
Jinzhou, China
ht0312wy@163.com

2nd Lei Liu
*College of Science, Liaoning University of Technology*
*name of organization (of Aff.)*
Jinzhou, China
liuleill@live.cn

3rd Yan-Jun Liu
*College of Science, Liaoning University of Technology*
*name of organization (of Aff.)*
Jinzhou, China
liuyanjun@live.com

*Abstract*—This paper considers the distributed nonlinear placement problem of a class of unmanned aerial vehicles (UAVs) multi-cluster systems. The goal is to determine the location of each cluster agent according to the constraints of the network topology. Specifically, the intermediate agents within each cluster are placed into the desired shape while minimizing the sum of the squares of the length of the Euclidean links between the center of each cluster and the members of the corresponding cluster. In detail, the intermediate agents within each cluster are placed into the desired shape while minimizing the sum of the squares of the length of the Euclidean links between the center of each cluster and the members of the corresponding cluster.

*Index Terms*—Distributed algorithm, Nash equilibrium(NE), unmanned aerial vehicles (UAVs), nonlinear placement

## I. INTRODUCTION

With the development of multi-disciplines such as computer science and artificial intelligence, nonlinear multi-agent systems, which have important application value in wireless sensor networks, military reconnaissance, intelligent transportation and other fields, have become a research hotspot in the field of control theory and control, and have achieved a lot of results, such as consistency control [1], non-cooperative game and so on. The nonlinear layout problem of multi-cluster system of UAVs is particularly concerned, which aims to determine some agent positions according to the constraints of network topology, so as to achieve the target formation and minimize the square sum of the Euclidean link lengths between each cluster centre and the corresponding cluster members. For example, the location of express warehouses needs to meet specific requirements while minimizing transportation costs.

Formation control of multiple UAVs systems has been a research hotspot for many years [2-3]. For the multiple-UAVs formation problem, the distributed control method has great advantages compared with the centralized method, mainly reflected in: 1) To achieve "decentralization", each agent can communicate with its neighbor agents in one-way or two-way communication, without the need for a central agent, thus saving communication costs. 2) Each agent can make individual calculation decisions and has good autonomy. 3) The system has good scalability and robustness. In practical applications, even if a single agent is destroyed, other agents can still complete the target task as usual. In recent years, in order to realize multi-agent autonomous formation, many scholars have proposed distributed algorithms. In [4], the novel open-loop Nash strategy design method is proposed, which enables each UAV to be implemented in a completely distributed manner by estimating its terminal state. In [5], the formation maintenance and reconfiguration of UAVs without obstacles and with obstacle avoidance characteristics are firstly studied. By designing the position and speed consistency control law between each UAV, each UAV and virtual leader, the swarm formation control and maintenance are realized. By changing the relative position relationship between each UAV and virtual leader, the formation transformation is realized. A collision prediction mechanism is introduced to determine whether each UAV needs to avoid obstacles. Then, on the basis of the above control law, combined with the artificial potential field method, the swarm formation control and obstacle avoidance maintenance are realized. Based on [5], a distributed obstacle avoidance controller was proposed in [6], which contains three detailed control items: obstacle avoidance item, collision avoidance item between robots and formation reconstruction. In practical application, an outdoor experimental platform with 6 quadrotors is built to verify the practicability of the proposed method. The results show that the three control conditions are compatible with each other, and the obstacle avoidance and formation control of the four-rotor aircraft can be realized.

In recent years, game theory has been widely used. For example, smart grids, mobile sensor networks, and communication networks. Distributed decision making of multi-agent systems based on game theory has been paid more and more attention by researchers. Nash equilibrium search problem is particularly concerned in game theory. Nash equilibrium is to find an optimal strategy that can maximize the agent's expected returns in a given environment. In any game, the

agent cannot obtain more returns by changing its actions. In [7], this paper studies the Nash equilibrium search problem in non-cooperative games in which the agent local objective function cannot be clearly expressed. As an alternative, the output value of the agent's local objective function should be measurable. An extreme value optimizer is designed to solve the Nash equilibrium of non-cooperative games. The design of extremum seeker is based on the dynamic mean agreement protocol and sinusoidal jitter signal modulation. On the basis of [7], Nian et al. studied the distributed Nash equilibrium of multi-cluster games in the switched communication topology in [8]. Specifically, the communication topology switches between a set of jointly linked directed graphs. Firstly, a new distributed game search algorithm is designed by using consensus protocol and gradient game rules. Then, in order to enable the agent to use only partial decision information, a more general game NE search algorithm is designed by using Leader-following control method, assuming that the switching topology between clusters is directed and strongly connected. Each agent intends to optimize its own objective function as a self-interested agent, but in reality, the agent coexists with competition. In [9], the problem of finding Nash equilibrium (NE) in a multi-agent system in a cooperation-competition network is studied. Due to the possibility of disconnection of the internal communication topology of the cluster, an algorithm using singular perturbation technique is designed, which divides the system into two different time scales. In high-speed systems, a new estimation algorithm is proposed to estimate the total cost function of disconnected subnets. In the slow system, NE search is based on gradient algorithm, and the convergence of the algorithm is analyzed by Lyapunov stability theory. The goal of each agent is to optimize a total cost function that takes into account both cooperative and non-cooperative interactions with other agents. Meanwhile, both its own interests and those of its collaborators are considered too.

Looking back on the above research results of multi-cluster game, this paper studies the nonlinear layout problem of multi-cluster UAVs. Since formation is automatically guaranteed by distributed method, it can be regarded as a special layout problem, so that the Euclidean length of each link is minimized. The formation control of multi-cluster unmanned aerial systems is modeled as a distributed nonlinear layout problem. The distributed nonlinear problem is equivalent to the cluster game problem to find the Nash equilibrium of the cluster game problem, and then the distributed observer method and gradient descent Nash equilibrium search algorithm are designed. Compared with previous algorithms, the biggest feature of our algorithm is that it can deal with time-varying NE, which makes the algorithm more difficult to design and converge. Meanwhile, our algorithm converges faster than previous algorithms.

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
