# OpenReview forum: "Distributed Nash Equilibrium Seeking for Multi-cluster UAVs Formation"
_IEEE.org/ICIST/2024/Conference — IEEE ICIST 2024 Conference Submission_

### Official Review · Reviewer_YQ5Q · 2024-08-21
**Lack of innovation**

**Rating:** 2
**Confidence:** 5

**Review:**

This article lacks innovation. And the writing structure of the article is not clear.

---

### Official Review · Reviewer_Ugxf · 2024-08-23
**Lack of innovation.**

**Rating:** 2
**Confidence:** 5

**Review:**

In general, there is a lack of innovation. The overall structure is confused and not up to the qualified level, so we decided to reject the manuscript.

---

### Official Review · Reviewer_xbnC · 2024-08-24
**Review Comments for Manuscript NO.179**

**Rating:** 3
**Confidence:** 4

**Review:**

The manuscript provides an abstract and introduction but lacks substantial content in other critical sections, including theoretical explanations, detailed methodologies, and simulation results. A manuscript is expected to present a complete and thorough exploration of the research topic, including problem formulation, methodology, results, and discussion.
Without these essential components, the manuscript fails to provide a comprehensive understanding of the research conducted and its contributions to the field. Therefore, the manuscript, in its current form, is incomplete and does not meet the standards required for publication.

---

### Decision · Program_Chairs · 2024-09-08

Accept (Oral)